# LncRNA MEG3 Regulates *Glaesserella parasuis*-Induced Apoptosis of Porcine Alveolar Macrophages via Regulating ssc-miR-135/CASP8 Axis

**DOI:** 10.3390/microorganisms13102287

**Published:** 2025-10-01

**Authors:** Yongchao Jia, Meiling Qian, Xinlu Sun, Ronglan Yin, Na Li, Aobo Shen, Haoran Wang, Fanhua Zeng, Yuanyuan Zhou, Ronghuan Yin

**Affiliations:** 1Key Laboratory of Livestock Infectious Diseases, Ministry of Education, and Key Laboratory of Ruminant Infectious Disease Prevention and Control (East), Ministry of Agriculture and Rural Affairs, College of Animal Science and Veterinary Medicine, Shenyang Agricultural University, 120 Dongling Road, Shenyang 110866, China; jiayongchao0320@163.com (Y.J.); qianmeiling0124@163.com (M.Q.); 18247530810@163.com (X.S.); lina001028@163.com (N.L.); 15642032160@163.com (A.S.); 15566446123@163.com (H.W.); 15541965306@163.com (F.Z.); zhouyy26@syau.edu.cn (Y.Z.); 2State Key Laboratory for Diagnosis and Treatment of Severe Zoonotic Infectious Diseases, Key Laboratory of Zoonosis Research, Ministry of Education, Jilin University, Changchun 130012, China; yinronglan@163.com

**Keywords:** LncRNA MEG3, ssc-miR-135, CASP8, *Glaesserella parasuis*, apoptosis

## Abstract

*Glaesserella parasuis* (*G. parasuis*), a common pathogenic bacterium in the porcine respiratory tract, can cause porcine polyserositis, arthritis, and meningitis. Alveolar macrophages are the first line of defense in the pulmonary innate immunity, and their abnormal apoptosis plays a critical role in the pathogenic process of *G. parasuis*. Long non-coding RNA maternally expressed gene 3 (MEG3) is associated with *G. parasuis* infection, but its mechanism remains incompletely unclear. This study aimed to investigate the role of MEG3 in *G. parasuis*-induced apoptosis of the porcine alveolar macrophage cell line 3D4/21 and its detailed molecular mechanism. Here, we found that MEG3 overexpression promoted *G. parasuis*-induced apoptosis and upregulated key extrinsic pathway proteins caspase-8 (CASP8) and caspase-3 (CASP3). Mechanistically, MEG3 functioned as a competing endogenous RNA by sponging ssc-miR-135, which directly targets and inhibits CASP8. Consequently, MEG3 overexpression alleviated ssc-miR-135-mediated repression of CASP8. Functional rescue experiments confirmed that either ssc-miR-135 mimic or CASP8 siRNA reversed the pro-apoptotic effect of MEG3. In conclusion, this study reveals that MEG3 relieves the inhibitory effect of ssc-miR-135 on CASP8 through competitively binding, thereby regulating *G. parasuis*-induced apoptosis of 3D4/21 cells. This study provides new insights into the pathogenic molecular mechanism of *G. parasuis*.

## 1. Introduction

*Glaesserella parasuis* (*G. parasuis*), a Gram-negative opportunistic pathogen, mainly infects the respiratory system of pigs and can cause Glässer’s disease characterized by fibrinous polyserositis, arthritis, and meningitis, resulting in severe economic losses to the global swine industry [1,2,3,4]. Recent studies have shown that the pathogenic mechanism of *G. parasuis* not only involves direct damage to host cells by virulence factors but is also closely related to abnormal biological behaviors of host cells [5]. Among them, the abnormal regulation of apoptosis is a key link in *G. parasuis* breaking the host immune barrier and promoting infection spread [6]. Li et al. demonstrated that *G. parasuis* can induce P53-dependent cell death [7]. In addition, Yan et al. reported that *G. parasuis* can induce more significant apoptosis by inhibiting the expression of the anti-cell death gene *BCL2* [6]. A recent study indicated that *G. parasuis* can trigger endoplasmic reticulum (ER) stress through the PERK/eIF2α/ATF4/CHOP pathway, thereby leading to mitochondrial damage [8]. Although these studies have emphasized the pathways of *G. parasuis*-induced apoptosis, the specific molecular mechanisms and targets remain incompletely elucidated.

Long non-coding RNAs (lncRNAs), a class of non-coding RNA molecules longer than 200 nucleotides (nt), do not have protein-coding functions but are widely involved in biological processes such as cell apoptosis, proliferation, differentiation, and immune response through mechanisms including epigenetic regulation, post-transcriptional regulation, and acting as molecular sponges. To date, numerous studies have reported that lncRNAs can act as competitive endogenous RNAs (ceRNAs) to inhibit miRNA functions and regulate mRNA expression by competitively binding to miRNAs [9]. For example, lncRNA CASC2 can act as a “sponge” for miR-24 and miR-221, regulating caspase-3/8 (CASP3/8) to affect TNF-related apoptosis-inducing ligand (TRAIL) resistance in hepatocellular carcinoma [10]. LncRNA THUMPD3-AS1 inhibits apoptosis of ovarian cancer cells by regulating the miR-320d/ARF1 axis [11]. As a widely studied tumor suppressor-related lncRNA, lncRNA maternally expressed gene 3 (MEG3) has been confirmed to regulate cell apoptosis, proliferation, and other processes by targeting different miRNAs [12,13,14,15,16,17,18]. It has been reported that MEG3 affects the proliferation and apoptosis of psoriatic epidermal cells by targeting miR-21/CASP8 [19]. Silencing of MEG3 exacerbates lipopolysaccharide-stimulated apoptosis of human lung cells by regulating miR-4262 [20]. Previous studies have shown that MEG3 is involved in the inflammatory response and apoptosis of porcine alveolar macrophages infected with *G. parasuis* by regulating the miR-210/TLR4 axis [21]. However, the role and potential molecular mechanism of MEG3 in *G. parasuis*-induced apoptosis remain largely unknown. In this study, we constructed a novel MEG3-miRNA-mRNA ceRNA network regulatory axis to further explore the function and potential molecular mechanism of MEG3 in *G. parasuis*-induced apoptosis. The results of this finding enrich the research content on the pathogenic mechanism of *G. parasuis*.

## 2. Materials and Methods

### 2.1. Bacterial Strain and Cell Line

The highly virulent *G. parasuis* serovar 13 strain CY1201 was isolated from a diseased pig in Liaoning, China, and cultured at 37 °C on tryptic soy agar (TSA, Solarbio, Beijing, China) or in tryptic soy broth (TSB, Solarbio, Beijing, China) supplemented with 2% nicotinamide adenine dinucleotide (NAD, Solarbio, Beijing, China) and 5% fetal bovine serum (Sangon Biotech, Shanghai, China). The porcine alveolar macrophage cell line 3D4/21 (Newgainbio, Wuxi, China) was cultured in RPMI 1640 medium (Sangon Biotech, Shanghai, China) supplemented with 10% fetal bovine serum (Sangon Biotech, Shanghai, China), penicillin–streptomycin mixture (Servicebio, Wuhan, China), and non-essential amino acids (Sangon Biotech, Shanghai, China) at 37 °C in a 5% CO_2_ atmosphere.

### 2.2. Cell Transfection and Bacterial Infection

The MEG3 overexpression plasmid (OE-MEG3) was synthesized by BGI Genomics Co., Ltd. (Beijing, China), and the negative control pcDNA3.1 (+) was used as the control vector (OE-NC). ssc-miR-135 mimic (5′-UAUGGCUUUUUAUUCCUAUGUGA ACAUAGGAAUAAAAAGCCAUAUU-3′), mimic mock (Mimic NC, 5′-UUCUCCGAACGUGUCACGUTT ACGUGACACGUUCGGAGAATT-3′), ssc-miR-135 inhibitor (5′-UCACAUAGGAAUAAAAAGCCAUA-3′), inhibitor mock (Inhibitor NC inhibitor, 5′-CAGUACUUUUGUGUAGUACAA-3′), Silencer^®^ CASP8 siRNA (si-CASP8, 5′-GAAAGUGCUUCGGAUGAAUTT AUUCAUCCGAAGCACUUUCTT-3′) and Silencer^®^ Negative Control (si-NC, 5′-UUCUCCGAACGUGUCACGUTT ACGUGACACGUUCGGAGAATT-3′) were designed and synthesized by Genepharma (Shanghai, China). Plasmids were transfected using PolyFast Transfection Reagent (MCE, Monmouth Junction, NJ, USA), while miRNA mimics, miRNA inhibitors, and siRNAs were transfected using GP-transfect-Mate (Genepharma, Shanghai, China) according to the manufacturer’s instructions. Transient transfection was performed when cell confluency reached 70–90%. After 8 h of transfection, the medium was replaced, and cells were cultured for 24 or 36 h. Cells were infected with *G. parasuis* for which multiplicity of infection (MOI) was 20 for 24 h, followed by further detection or RNA/protein extraction.

### 2.3. Cell Viability Assay

Cell viability was detected using the Cell Counting Kit-8 (CCK-8, Jiangsu Cowin Biotech Co., Ltd., Taizhou, China) according to the manufacturer’s instructions. Briefly, cells were seeded into 96-well plates (Sangon Biotech, Shanghai, China) at a density of 2000 cells per well in 100 μL of medium. After 24 h of transfection, the cells were treated with *G. parasuis* for 24 h. Subsequently, 10 μL of CCK-8 solution was added to each well, and the cells were incubated for an additional 2 h. The absorbance at 450 nm was measured using a microplate reader (Servicebio, Wuhan, China).

### 2.4. Annexin V-FITC/PI Staining Assay

According to the manufacturer’s instructions, 5 × 10^5^ cells were incubated with Annexin V-FITC Reagent and PI Reagent (Annexin V-FITC/PI Cell Apoptosis Detection Kit, Elabscience Biotechnology, Wuhan, China) at room temperature in the dark for 20 min, and the apoptosis rate was detected using BD FACSAria III (Becton Dickinson, Franklin Lakes, NJ, USA).

### 2.5. Bioinformatics Analysis

RNAhybrid 2.1.2 predicts potential binding sites between lncRNA, miRNA, and mRNA. The prediction of miRNA target genes was performed via OmicStudio tools (LC-Bio Technologies Co., Ltd., Hangzhou, China; https://www.omicstudio.cn/tool/, accessed on 20 July 2025), which integrates two algorithms: TargetScan (v5.0) and miRanda (v3.3a). The screening thresholds were set as follows: TargetScan_score ≥ 50 and miRanda_Energy < −10. To improve the reliability of predictions, the target genes identified by both algorithms were combined, and their overlapping genes were retained for subsequent analysis. Gene Ontology (GO) terms/Kyoto Encyclopedia of Genes and Genomes (KEGG) enrichment analysis provides all GO terms/KEGG pathways that are significantly enriched in miRNA target genes compared to the genome background. All miRNA target genes were mapped to GO terms/KEGG pathways in the GO database (http://www.geneontology.org/, accessed on 20 July 2025) and KEGG database (https://www.kegg.jp/, accessed on 20 July 2025); gene numbers were calculated for every term/pathway; significantly enriched GO terms/KEGG pathways in miRNA target genes compared to the genome background were defined by hypergeometric test. Database for Annotation, Visualization and Integrated Discovery (DAVID, https://david.ncifcrf.gov/, accessed on 20 July 2025), Metascape (https://metascape.org/, accessed on 20 July 2025) and WebGestalt (https://www.webgestalt.org/, accessed on 20 July 2025) and “clusterProfiler” package were used for data analyses. GO database and KEGG database were used for enrichment analysis, with the species chosen being *Sus scrofa*. For the enrichment analysis, the criteria were set as follows: *p* value < 0.05 and enrichment fold (EF; ratio of target gene proportion in the pathway to background gene proportion in the pathway) > 1.5.

### 2.6. Dual-Luciferase Reporter Assay

RNAhybrid 2.1.2 was used to predict potential binding sites between MEG3 and ssc-miR-135, as well as between ssc-miR-135 and the 3′ untranslated region (UTR) of *CASP8*. The wild type MEG3 (MEG3-Wt), mutant type MEG3 (MEG3-Mut), wild type *CASP8* 3′ UTR (*CASP8* 3′ UTR-Wt), and mutant type *CASP8* 3′ UTR (*CASP8* 3′ UTR-Mut) sequences were cloned into the dual-luciferase reporter vector pmirGLO (Genepharma, Shanghai, China), which was synthesized by BGI Genomics Co., Ltd. (Shenzhen, China). The constructed dual-luciferase reporter vectors were co-transfected with miR-135 mimic into 3D4/21 cells. Luciferase activity was determined using the dual-Luciferase Kit (Jiangsu Cowin Biotech Co., Ltd., Taizhou, China) according to the manufacturer’s instructions.

### 2.7. RNA Extraction and Quantitative Real-Time PCR (qPCR) Detection

Total cellular RNA was extracted using the Trizol RNA Extraction Kit (Vazyme, Nanjing, China). First-strand cDNA was synthesized using a reverse transcription kit (Vazyme, Nanjing, China) according to the manufacturer’s instructions. cDNA for detecting the expression level of ssc-miR-135 was synthesized using the SuperStar miRNA First-Strand cDNA Synthesis Kit (by tailing A) (Jiangsu Cowin Biotech Co., Ltd., Taizhou, China). qPCR was performed to detect the expression levels of MEG3, ssc-miR-135, and mRNA. *GAPDH* and *U6* were used as internal reference genes for expression normalization. 2^−ΔΔCt^ method was used to analyze the experimental results. The primer sequences are shown in Table 1.

### 2.8. Western Blot

Total protein was extracted using RIPA lysis buffer (Epizyme, Shanghai, China) containing protease inhibitors and phosphatase inhibitors, and protein concentration was determined using a BCA protein assay kit (Epizyme, Shanghai, China). Equal amounts of protein were loaded onto 12.5% sodium dodecyl sulfate polyacrylamide gel electrophoresis (SDS-PAGE) (Epizyme, Shanghai, China) for separation, then transferred to 0.22 μm PVDF membranes (Abcholal, Wuhan, China). The membranes were blocked with 5% skimmed milk (Epizyme, Shanghai, China) at room temperature for 2 h, then incubated with primary antibodies anti-CASP3 (1:1000 dilution, #T40044, Abmart, Shanghai, China), anti-Cleaved-CASP3 (1:1000 dilution, #T40044, Abmart, Shanghai, China), anti-CASP8 (1:1000 dilution, #PK85346, Abmart, Shanghai, China), anti-Cleaved-CASP8 (1:1000 dilution, #PK85387, Abmart, Shanghai, China), and anti-GAPDH (1:10,000 dilution, #10494-1-AP, Proteintech Group, Wuhan, China) at 4 °C overnight. After washing, the membranes were incubated with horseradish peroxidase (HRP)-conjugated secondary antibody (1:8000 dilution, #LF108, Epizyme, Shanghai, China). Chemiluminescent signals were detected using an ECL kit (Ncmbio, Suzhou, China) and visualized using a gel imaging system (Servicebio, Wuhan, China). The expression of each protein was normalized to GAPDH, and semi-quantitative analysis was performed using ImageJ.

### 2.9. Statistical Analysis

Statistical analysis and visualization were conducted using SPSS 15.0 software (SPSS Inc., Chicago, IL, USA) and GraphPad Prism 8.0 software (GraphPad Software, San Diego, CA, USA). Data are presented as mean ± standard deviation (SD). Comparisons between two groups were made using an independent samples *t*-test, with a significance level set at *p* < 0.05, indicating a statistically significant difference.

## 3. Results

### 3.1. Overexpression of MEG3 Promotes G. parasuis-Induced Apoptosis of 3D4/21 Cells

To clarify the effect of MEG3 on *G. parasuis*-induced apoptosis of 3D4/21 cells, we constructed a MEG3 overexpression plasmid, transfected it into 3D4/21 cells, and then infected the cells with *G. parasuis*. First, qPCR was used to detect the overexpression efficiency. Compared with the control group, the expression level of MEG3 in the MEG3 overexpression group was significantly increased (*p* < 0.01), indicating the successful establishment of a MEG3 overexpressing cell model in *G. parasuis*-infected 3D4/21 cells (Figure 1A). The CCK-8 assay showed that overexpression of MEG3 significantly reduced cell viability (*p* < 0.01) (Figure 1B). Flow cytometry analysis revealed that overexpression of MEG3 significantly increased the apoptosis rate (*p* < 0.01) (Figure 1C,D). qPCR analysis of apoptosis-related genes showed that MEG3 overexpression promoted the mRNA expression of *FASLG*, *TNF*, *CASP8*, *BAX*, and *CASP3* while inhibiting the mRNA expression of *BCL2* (*p* < 0.05 or *p* < 0.01). In contrast, the mRNA levels of *TP53*, *CYCS*, *CASP9*, *AIFM1*, and *ENDOG* showed no significant changes (*p* > 0.05) (Figure 1E). Most of these differentially expressed genes are involved in the extrinsic apoptosis pathway, suggesting that MEG3 may be involved in the extrinsic apoptosis pathway. Furthermore, *G. parasuis* significantly increased the protein expression levels of Cleaved-CASP8/CASP8 and Cleaved-CASP3/CASP3 in 3D4/21 cells, and this change was exacerbated by MEG3 overexpression (*p* < 0.01) (Figure 1F).

As CASP8 is a key initiator of the extrinsic apoptosis pathway, to further confirm that MEG3 exerts its function through the extrinsic apoptosis pathway, we used Z-IETD-FMK treatment to inhibit its expression, thereby suppressing the activation of extrinsic apoptosis. First, qPCR was performed to detect the effects of different concentrations of Z-IETD-FMK on mRNA levels of *CASP8* and *CASP3* in *G. parasuis*-infected 3D4/21 cells. The results showed that, after *G. parasuis* infection of 3D4/21 cells, the mRNA levels of *CASP8* and *CASP3* were significantly increased (*p* < 0.01). After treatment with different concentrations of Z-IETD-FMK, the mRNA levels of *CASP8* and *CASP3* gradually decreased with increasing inhibitor concentration (*p* < 0.01) (Figure 2A), indicating that Z-IETD-FMK can effectively inhibit the activation of the extrinsic apoptosis pathway after *G. parasuis* infection. Furthermore, after transfection with the MEG3 overexpression plasmid, 3D4/21 cells were treated with 20 μM of Z-IETD-FMK followed by *G. parasuis* infection. qPCR results showed that the increase in mRNA levels of *CASP8* and *CASP3* caused by MEG3 overexpression was inhibited after the addition of Z-IETD-FMK (*p* < 0.01) (Figure 2B). Flow cytometry analysis revealed that the increase in apoptosis rate induced by MEG3 overexpression was also suppressed after Z-IETD-FMK treatment (*p* < 0.01) (Figure 2C,D). These results indicate that overexpression of MEG3 promotes *G. parasuis*-induced apoptosis of 3D4/21 cells through the extrinsic apoptosis pathway.

### 3.2. MEG3 as a Molecular Sponge for ssc-miR-135

It has been reported that MEG3 can bind to miR-135 [22]. Therefore, we identified ssc-miR-135 as a potential target, as it is highly conserved in mammals (Figure 3A). RNAhybrid prediction revealed a binding site between MEG3 and ssc-miR-135 (Figure 3B). Following further mutation of the binding site, in 3D4/21 cells transfected with ssc-miR-135 mimic, the luciferase activity of MEG3-Wt decreased (*p* < 0.01), while that of MEG3-Mut showed no significant effect (*p* > 0.05) (Figure 3C). The results indicate that MEG3 targets ssc-miR-135. Subsequently, we measured the expression changes in ssc-miR-135 in 3D4/21 cells infected with *G. parasuis*. The results showed that ssc-miR-135 exhibited time and dose-dependent downregulation (*p* < 0.05 or *p* < 0.01) (Figure 3D). We further examined the expression of ssc-miR-135 in *G. parasuis*-infected 3D4/21 cells with MEG3 overexpression. The results showed that overexpression of MEG3 significantly reduced the expression of ssc-miR-135 (*p* < 0.01) (Figure 3E). In summary, these results indicate that MEG3 directly binds to ssc-miR-135 in 3D4/21 cells.

### 3.3. ssc-miR-135 Regulates G. parasuis-Induced Apoptosis in 3D4/21 Cells

To clarify whether ssc-miR-135 plays a role in *G. parasuis*-induced apoptosis of 3D4/21 cells, we first transfected 3D4/21 cells with ssc-miR-135 mimic or ssc-miR-135 inhibitor, and verified the overexpression/knockdown efficiency (Figure 4A,G). Subsequently, *G. parasuis* infection was performed in 3D4/21 cells with ssc-miR-135 overexpression or knockdown. CCK-8 assay indicated that, compared with the control group, transfection with ssc-miR-135 mimic significantly increased cell viability, while transfection with ssc-miR-135 inhibitor significantly decreased cell viability (*p* < 0.01) (Figure 4B,H). To determine whether the reduction in cell viability was attributed to apoptosis, flow cytometry analysis was performed. The results revealed that transfection with ssc-miR-135 mimic significantly reduced the apoptosis rate of 3D4/21 cells induced by *G. parasuis* infection, whereas transfection with ssc-miR-135 inhibitor increased the apoptosis rate (*p* < 0.01) (Figure 4C,D,I,J). qPCR analysis revealed that transfection with ssc-miR-135 mimic significantly inhibited the expression of *CASP8* and *CASP3* mRNA levels induced by *G. parasuis* infection, while an opposite trend was observed in cells transfected with ssc-miR-135 inhibitor (*p* < 0.01) (Figure 4E,K). Western blot analysis indicated that, compared with the control group, the protein expression ratios of Cleaved-CASP8/CASP8 and Cleaved-CASP3/CASP3 were decreased by transfection with ssc-miR-135 mimic and increased by transfection with ssc-miR-135 inhibitor (*p* < 0.01) (Figure 4F,L). Collectively, these results indicate that ssc-miR-135 negatively regulates *G. parasuis*-induced apoptosis in 3D4/21 cells.

### 3.4. ssc-miR-135 Directly Interacts with CASP8

To further explore the interaction between miRNA and its downstream targets, we used TargetScan and miRanda to predict the target genes of ssc-miR-135 (Figure 5A) (prediction results are shown in Appendix A). GO and KEGG enrichment analysis of the predicted target genes revealed enrichment in GO terms such as cell death, apoptotic signaling pathway, and apoptotic process, as well as the Apoptosis KEGG pathway (Figure 5B,C) (enrichment results are shown in Appendix A). RNAhybrid prediction identified a binding site between ssc-miR-135 and the 3′ UTR of *CASP8* (Figure 5D). Subsequent dual-luciferase reporter assay results showed that, when ssc-miR-135 mimic were co-transfected with the *CASP8* 3′ UTR-Wt, luciferase activity was significantly reduced (*p* < 0.01), while there was no significant change in the *CASP8* 3′ UTR-Mut (*p* > 0.01) (Figure 5E). These findings indicate that ssc-miR-135 targets CASP8. Additionally, previous results showed that ssc-miR-135 mimic significantly reduced *CASP8* mRNA levels and Cleaved-CASP8/CASP8 protein levels, while ssc-miR-135 interference had the opposite trend (Figure 4). Collectively, these results indicate that CASP8 is a target gene of ssc-miR-135.

### 3.5. MEG3 Regulates G. parasuis-Induced Apoptosis in 3D4/21 Cells via ssc-miR-135/CASP8 Axis

Given that MEG3 can regulate *G. parasuis*-induced apoptosis in 3D4/21 cells and interacts with ssc-miR-135, while ssc-miR-135 negatively regulates *G. parasuis*-induced apoptosis in 3D4/21 cells, we hypothesized that MEG3 regulates the expression of CASP8 via ssc-miR-135, forming a ceRNA regulatory relationship (Figure 5F). To verify this hypothesis, 3D4/21 cells were co-transfected with MEG3 overexpression plasmid and ssc-miR-135 mimic, followed by infection with *G. parasuis*. CCK-8 assays revealed that ssc-miR-135 mimic reversed the reduction in cell viability caused by MEG3 overexpression (*p* < 0.01) (Figure 6A). Additionally, we found that, in *G. parasuis*-infected 3D4/21 cells, co-transfection of MEG3 overexpression plasmid and ssc-miR-135 mimic significantly reduced apoptosis rates compared to transfection with MEG3 overexpression plasmid alone (*p* < 0.01) (Figure 6B,C). qPCR results showed that co-transfection of MEG3 overexpression plasmid and ssc-miR-135 mimic significantly increased ssc-miR-135 expression (*p* < 0.01) (Figure 6E) and decreased MEG3 and CASP8 expression (*p* < 0.01) (Figure 6D,F) compared to transfection with MEG3 overexpression plasmid alone. Western blot analysis indicated that, compared with transfection with MEG3 overexpression plasmid alone, co-transfection with MEG3 overexpression plasmid and ssc-miR-135 mimic significantly reduced the protein expression of Cleaved-CASP8/CASP8 (*p* < 0.01) (Figure 6G). These results indicate that MEG3 acts as a molecular sponge for ssc-miR-135 to regulate the expression of CASP8.

Finally, to clarify that MEG3 regulates 3D4/21 cell apoptosis via ssc-miR-135/CASP8 axis, we first designed siRNA targeting CASP8 and verified its knockdown efficiency (Figure 7A). Subsequently, 3D4/21 cells were co-transfected with MEG3 overexpression plasmid and ssc-miR-135 mimic, as well as with MEG3 overexpression plasmid and si-CASP8, respectively. CCK-8 assay results demonstrated that ssc-miR-135 mimic reversed the decrease in cell viability caused by MEG3 overexpression, and a similar reversing effect was observed upon CASP8 knockdown (*p* < 0.01) (Figure 7B). Additionally, we found that, in *G. parasuis*-infected 3D4/21 cells, compared to transfection with MEG3 overexpression plasmid alone, co-transfection with MEG3 overexpression plasmid and ssc-miR-135 mimic, as well as co-transfection with MEG3 overexpression plasmid and si-CASP8, significantly reduced the apoptosis rate (*p* < 0.01) (Figure 7C,D). qPCR and Western blot analysis results showed that, compared to transfection with MEG3 overexpression plasmid alone, co-transfection with MEG3 overexpression plasmid and ssc-miR-135 mimic, as well as co-transfection with MEG3 overexpression plasmid and si-CASP8, significantly reduced the expression of CASP3 (*p* < 0.01) (Figure 7E,F). In summary, these findings indicate that MEG3 regulates *G. parasuis*-induced apoptosis in 3D4/21 cells via ssc-miR-135/CASP8 axis.

## 4. Discussion

Glässer’s disease caused by *G. parasuis* has caused serious economic losses to the pig industry [23]. It has been reported that many lncRNAs with differential expression during *G. parasuis* infection have been identified, but their specific functions remain unclear [24]. Beyond its well-known role as a tumor-suppressive lncRNA, mounting evidence indicates that MEG3 plays crucial regulatory roles in the occurrence and development of various diseases. In this study, we found that overexpression of MEG3 can significantly reduce the viability of 3D4/21 cells infected with *G. parasuis* and increase the apoptosis rate, which is consistent with previous research findings [21]. At the same time, we further observed that overexpression of MEG3 leads to a significant increase in the mRNA and protein levels of apoptosis-initiating factors and apoptosis-effecting factors CASP8 and CASP3 in the extrinsic apoptosis pathway. These findings indicate that MEG3 may regulate *G. parasuis*-induced apoptosis in 3D4/21 cells through the extrinsic apoptotic pathway. Our results further clarify the role of MEG3 in *G. parasuis*-induced apoptosis in 3D4/21 cells. Accumulating research evidence has demonstrated that lncRNAs are associated with host–pathogen interactions [25,26,27,28,29]. Previous studies have confirmed that the cytolethal distending toxin (CDT), a virulence factor of *G. parasuis*, can enhance the adhesion and invasion of *G. parasuis* to host cells, activate the DNA damage response, and induce cell cycle arrest and host cell apoptosis [7,30,31]. However, whether the apoptosis regulated by MEG3 involves virulence factors of *G. parasuis* remains to be further studied.

Given that ssc-miR-135 is differentially expressed after *G. parasuis* infection [32], and MEG3 can promote bovine myoblast differentiation through the sponge miR-135 [22], we further explored the function and interaction of MEG3 and ssc-miR-135 in the process of *G. parasuis*-induced apoptosis in 3D4/21 cells. In our study, we again confirmed that ssc-miR-135 is a target miRNA of MEG3 using a dual-luciferase reporter assay. ssc-miR-135 is downregulated in 3D4/21 cells infected with *G. parasuis*, and knockdown of ssc-miR-135 promotes *G. parasuis*-induced apoptosis in 3D4/21 cells, which is consistent with the effect of MEG3 overexpression. Meanwhile, overexpression of ssc-miR-135 reverses the promotional effect of MEG3 overexpression on *G. parasuis*-induced apoptosis in 3D4/21 cells. Our study first revealed that MEG3 can regulate *G. parasuis*-induced apoptosis in 3D4/21 cells through competitively binding to ssc-miR-135.

MicroRNAs (miRNAs) exert their biological functions by post-transcriptionally regulating their downstream targets [33]. Yu et al. reported that miR-135 regulates NLRP3 inflammasome activation by inhibiting the expression of CaSR [34]. Previous studies have indicated that ssc-miR-135 may promote *G. parasuis*-induced inflammatory responses through the NF-κB signaling pathway [32]. However, whether ssc-miR-135 regulates the pathogenic process of *G. parasuis* infection by binding to downstream targets remains unknown. In this study, we first identified CASP8 as a downstream target of ssc-miR-135 and also as a downstream effector molecule of the MEG3/CASP8 axis. Cleavage of CASP8 is critical for the activation of the extrinsic apoptotic pathway, which ultimately induces the cleavage of CASP3 protein [35]. Our findings indicate that ssc-miR-135 participates in regulating *G. parasuis*-induced apoptosis in 3D4/21 cells by negatively regulating CASP8 expression. Additionally, LncRNA can act as a miRNA sponge and weaken mRNA regulation [36]. Similarly, we found that CASP8 expression is promoted by MEG3 overexpression, while ssc-miR-135 overexpression eliminates this effect. Furthermore, when CASP8 is inhibited, the promotion of downstream CASP3 expression caused by MEG3 overexpression is suppressed, thereby attenuating *G. parasuis*-induced apoptosis in 3D4/21 cells. These results reveal a novel regulatory network mechanism by which MEG3 regulates *G. parasuis*-induced apoptosis in 3D4/21 cells via the ssc-miR-135/CASP8 axis (Figure 8).

This study has several limitations. First, it remains unclear whether other miRNAs are involved in the process by which MEG3 regulates *G. parasuis*-induced apoptosis. Moreover, the in vitro porcine alveolar macrophage model can only simulate local infection, and the regulatory mechanism of the MEG3/ssc-miR-135/CASP8 axis during systemic infection requires further evaluation in animal models or using different cell lines.

## 5. Conclusions

This study demonstrates that MEG3 can regulate *G. parasuis*-induced apoptosis in 3D4/21 cells via ssc-miR-135/CASP8 axis. These findings provide new insights into the molecular mechanisms underlying the pathogenicity of *G. parasuis*.

## Figures and Tables

**Figure 1 microorganisms-13-02287-f001:**
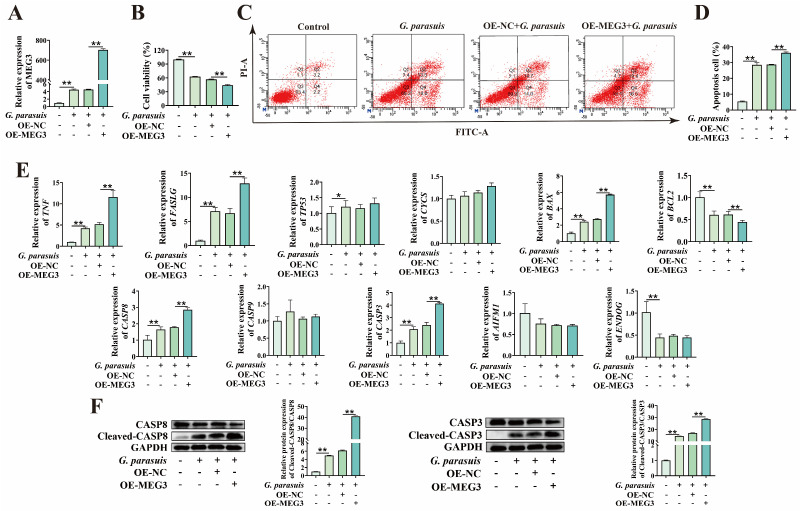
Effects of MEG3 overexpression on *G. parasuis*-induced apoptosis of 3D4/21 cells. (**A**) After transfection with the MEG3 overexpression plasmid, the expression of MEG3 in *G. parasuis*-infected 3D4/21 cells was detected by qPCR. “**” indicates *p* < 0.01. (**B**) The effect of MEG3 overexpression on cell viability of *G. parasuis*-infected 3D4/21 cells was detected by the CCK-8 assay. “**” indicates *p* < 0.01. (**C**) After transfection with the MEG3 overexpression plasmid, 3D4/21 cells were infected with *G. parasuis*, and the apoptosis rate was detected by flow cytometry. (**D**) Statistical analysis results of the apoptosis rate. “**” indicates *p* < 0.01. (**E**) After transfection with the MEG3 overexpression plasmid, mRNA expression levels of apoptosis-related genes in *G. parasuis*-infected 3D4/21 cells were detected by qPCR. “*” indicates *p* < 0.05; “**” indicates *p* < 0.01. (**F**) After transfection with the MEG3 overexpression plasmid, the protein expression levels of apoptosis-related genes in *G. parasuis*-infected 3D4/21 cells were detected by Western blot. “**” indicates *p* < 0.01.

**Figure 2 microorganisms-13-02287-f002:**
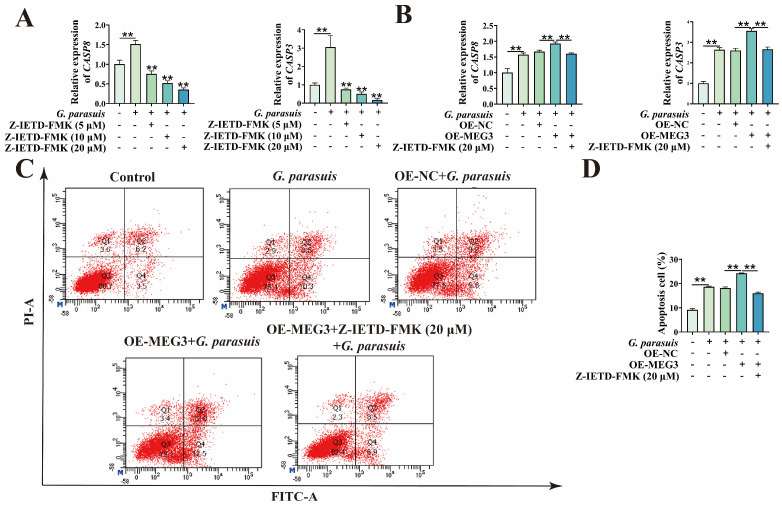
Effects of MEG3 overexpression on the extrinsic apoptosis pathway in *G. parasuis*-induced 3D4/21 cells. (**A**) After treatment with different concentrations of Z-IETD-FMK, the mRNA expression levels of apoptosis-related genes in *G. parasuis*-infected 3D4/21 cells were detected by qPCR. “**” indicates *p* < 0.01. (**B**) After transfection with the MEG3 overexpression vector and treatment with 20 μM of Z-IETD-FMK, the mRNA expression levels of apoptosis-related genes in *G. parasuis*-infected 3D4/21 cells were detected by qPCR. “**” indicates *p* < 0.01. (**C**) After transfection with the MEG3 overexpression vector and treatment with 20 μM of Z-IETD-FMK, the apoptosis rate was detected by flow cytometry. (**D**) Statistical analysis results of the apoptosis rate. “**” indicates *p* < 0.01.

**Figure 3 microorganisms-13-02287-f003:**
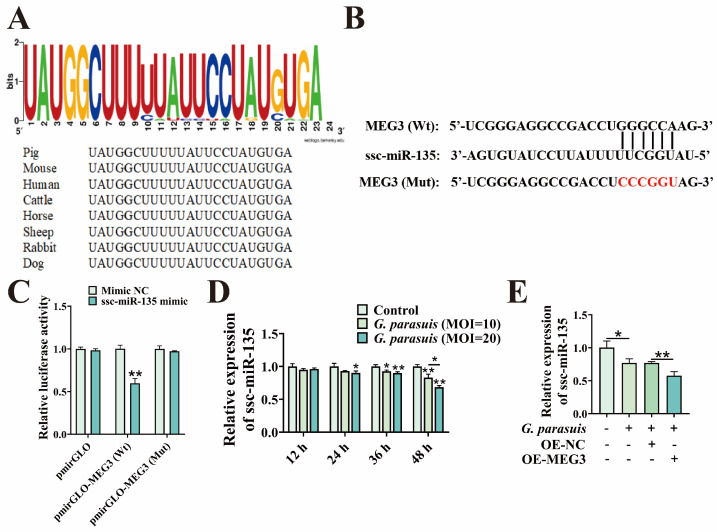
MEG3 targets ssc-miR-135. (**A**) Conservation analysis of ssc-miR-135 (visualized using the WEBLOGO online website). (**B**) MEG3 and ssc-miR-135 binding site map. Red represents the base after mutation. (**C**) Effect of co-transfection of MEG3-Wt/MEG3-Mut dual-luciferase reporter vectors with ssc-miR-135 mimic on relative luciferase activity in 3D4/21 cells. “**” indicates *p* < 0.01. (**D**) qPCR detection of ssc-miR-135 expression in *G. parasuis*-infected 3D4/21 cells. “*” indicates *p* < 0.05; “**” indicates *p* < 0.01. (**E**) qPCR detection of ssc-miR-135 expression in *G. parasuis*-infected 3D4/21 cells after transfection with the MEG3 overexpression vector. “*” indicates *p* < 0.05; “**” indicates *p* < 0.01.

**Figure 4 microorganisms-13-02287-f004:**
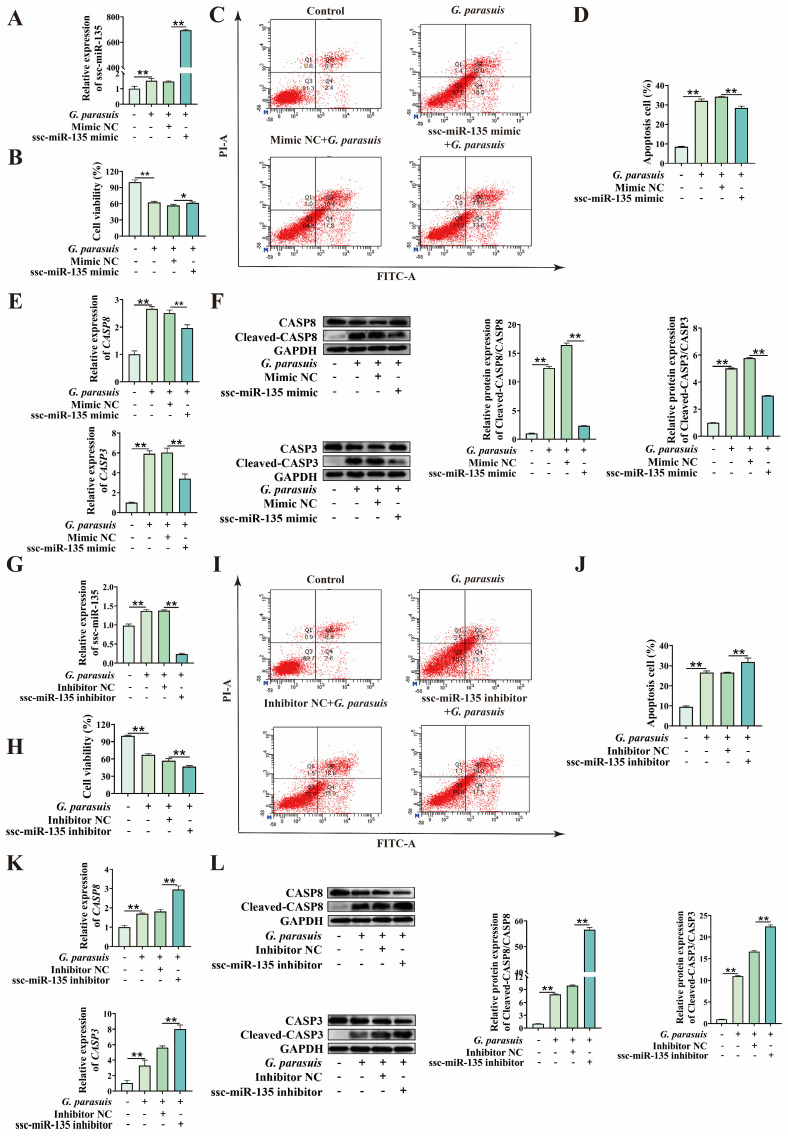
Effect of ssc-miR-135 on *G. parasuis*-induced apoptosis in 3D4/21 cells. (**A**) After transfection with ssc-miR-135 mimic, cells were infected with *G. parasuis*, and qPCR was used to detect the expression of ssc-miR-135 in *G. parasuis*-infected 3D4/21 cells. “**” indicates *p* < 0.01. (**B**) CCK-8 assay to assess the effect of ssc-miR-135 mimic on cell viability in *G. parasuis*-infected 3D4/21 cells. “*” indicates *p* < 0.05; “**” indicates *p* < 0.01. (**C**) After transfecting 3D4/21 cells with ssc-miR-135 mimic, cells were infected with *G. parasuis*, and apoptosis rates were measured by flow cytometry. (**D**) Statistical analysis results of apoptosis rate. “**” indicates *p* < 0.01. (**E**) qPCR detection of mRNA expression levels of apoptosis-related genes in *G. parasuis*-infected 3D4/21 cells after transfection with ssc-miR-135 mimic. “**” indicates *p* < 0.01. (**F**) After transfection with ssc-miR-135 mimic, Western Blot was used to detect the expression levels of apoptosis-related gene proteins in *G. parasuis*-infected 3D4/21 cells. “**” indicates *p* < 0.01. (**G**) After transfection with ssc-miR-135 inhibitor, cells were infected with *G. parasuis*, and qPCR was used to detect the expression of ssc-miR-135 in *G. parasuis*-infected 3D4/21 cells. “**” indicates *p* < 0.01. (**H**) CCK-8 assay to assess the effect of ssc-miR-135 inhibitor on cell viability in *G. parasuis*-infected 3D4/21 cells. “**” indicates *p* < 0.01. (**I**) After transfecting 3D4/21 cells with ssc-miR-135 inhibitor, cells were infected with *G. parasuis*, and apoptosis rates were measured by flow cytometry. (**J**) Statistical analysis results of apoptosis rate. “**” indicates *p* < 0.01. (**K**) qPCR detection of mRNA expression levels of apoptosis-related genes in *G. parasuis*-infected 3D4/21 cells after transfection with ssc-miR-135 inhibitor. “**” indicates *p* < 0.01. (**L**) After transfection with ssc-miR-135 inhibitor, Western Blot was used to detect the expression levels of apoptosis-related gene proteins in *G. parasuis*-infected 3D4/21 cells. “**” indicates *p* < 0.01.

**Figure 5 microorganisms-13-02287-f005:**
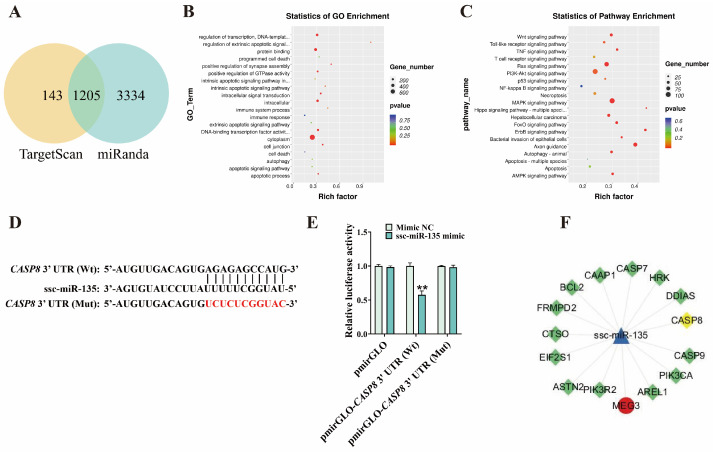
ssc-miR-135 targets CASP8. (**A**) Venn diagram showing the target genes of ssc-miR-135 from the TargetScan and miRanda databases. (**B**) Results of GO enrichment analysis for ssc-miR-135 target genes. (**C**) Results of KEGG enrichment analysis for ssc-miR-135 target genes. (**D**) Schematic diagram of the binding site between ssc-miR-135 and CASP8. Red represents the base after mutation. (**E**) Effect of co-transfection with *CASP8* 3′ UTR-Wt/*CASP8* 3′ UTR-Mut dual-luciferase reporter vectors and ssc-miR-135 mimic on relative luciferase activity in 3D4/21 cells. “**” indicates *p* < 0.01. (**F**) Schematic diagram of ceRNA network.

**Figure 6 microorganisms-13-02287-f006:**
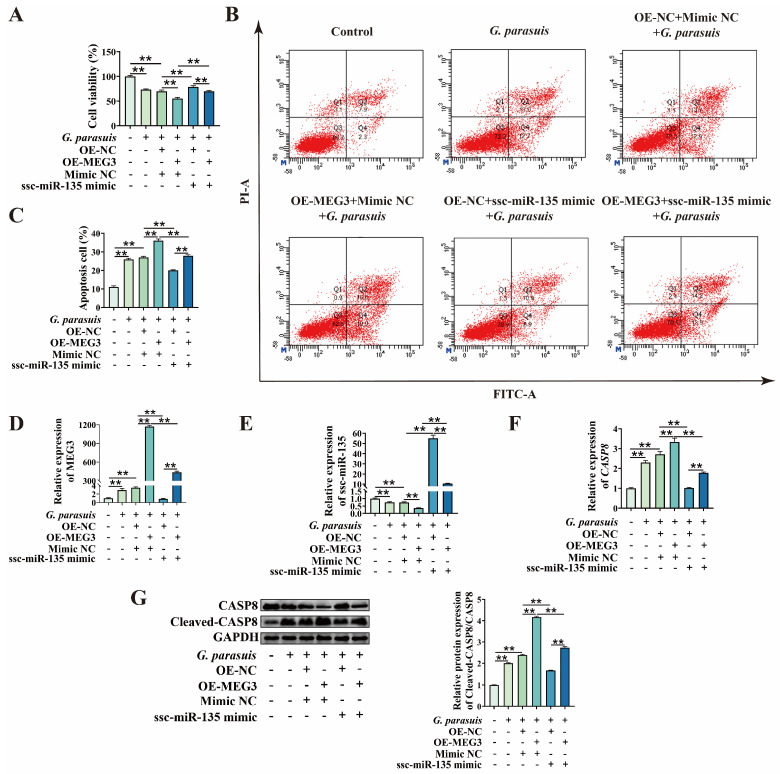
Effect of MEG3 on CASP8 expression via ssc-miR-135-mediated pathway. (**A**) CCK-8 assay was performed to detect the effect of co-transfection of MEG3 overexpression plasmid and ssc-miR-135 mimic on cell viability after *G. parasuis* infection of 3D4/21 cells. “**” indicates *p* < 0.01. (**B**) After co-transfection with MEG3 overexpression plasmid and ssc-miR-135 mimic, 3D4/21 cells were infected with *G. parasuis*, and apoptosis rates were detected by flow cytometry. (**C**) Statistical analysis results of apoptosis rate. “**” indicates *p* < 0.01. (**D**) After co-transfection of MEG3 overexpression plasmid and ssc-miR-135 mimic, qPCR was used to detect the expression of MEG3 and (**E**) ssc-miR-135 in *G. parasuis*-infected 3D4/21 cells. “**” indicates *p* < 0.01. (**F**) After co-transfection with MEG3 overexpression plasmid and ssc-miR-135 mimic, qPCR was performed to determine the mRNA level of *CASP8* in *G. parasuis*-infected 3D4/21 cells. “**” indicates *p* < 0.01. (**G**) After co-transfection of the MEG3 overexpression plasmid and ssc-miR-135 mimic, Western Blot analysis was conducted to detect the protein level of CASP8 in *G. parasuis*-infected 3D4/21 cells. “**” indicates *p* < 0.01.

**Figure 7 microorganisms-13-02287-f007:**
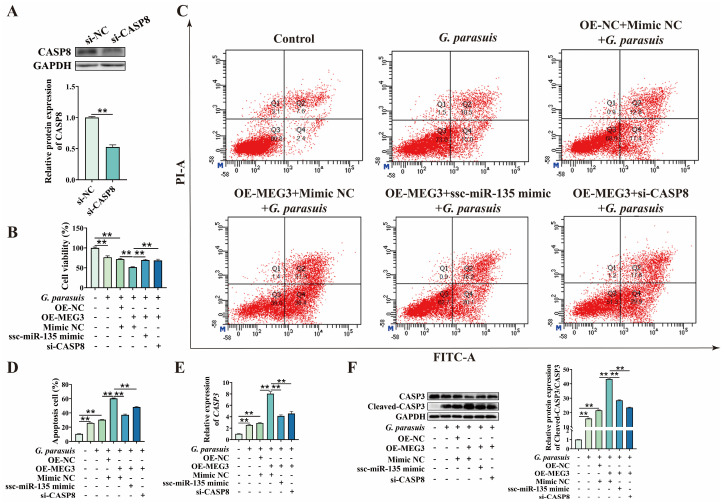
Functional verification of MEG3/ssc-miR-135/CASP8 axis in regulating *G. parasuis*-induced apoptosis in 3D4/21 cells. (**A**) Verification of CASP8 knockdown efficiency. “**” indicates *p* < 0.01. (**B**) CCK-8 assay was performed to detect the effect of co-transfection of MEG3 overexpression plasmid and ssc-miR-135 mimic, as well as MEG3 overexpression plasmid and si-CASP8, on cell viability after *G. parasuis* infection of 3D4/21 cells. “**” indicates *p* < 0.01. (**C**) After co-transfection with MEG3 overexpression plasmid and ssc-miR-135 mimic, as well as MEG3 overexpression plasmid and si-CASP8, 3D4/21 cells were infected with *G. parasuis*, and apoptosis rates were detected by flow cytometry. (**D**) Statistical analysis results of apoptosis rate. “**” indicates *p* < 0.01. (**E**) After co-transfection with MEG3 overexpression plasmid and ssc-miR-135 mimic, as well as MEG3 overexpression plasmid and si-CASP8, qPCR was performed to determine the mRNA level of *CASP3* in *G. parasuis*-infected 3D4/21 cells. “**” indicates *p* < 0.01. (**F**) After co-transfection with MEG3 overexpression plasmid and ssc-miR-135 mimic, as well as MEG3 overexpression plasmid and si-CASP8, Western Blot analysis was conducted to detect the protein level of CASP3 in *G. parasuis*-infected 3D4/21 cells. “**” indicates *p* < 0.01.

**Figure 8 microorganisms-13-02287-f008:**
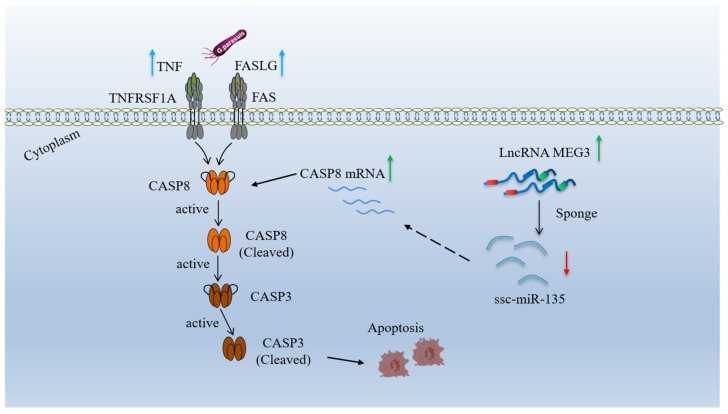
Schematic diagram of MEG3 regulates *G. parasuis*-induced apoptosis of porcine alveolar macrophages via regulating ssc-miR-135/CASP8 Axis. Red arrow: active.

**Table 1 microorganisms-13-02287-t001:** Primer sequences used for qPCR.

Genes	Primer Sequences (5′–3′)	Annealing Temperature (°C)	Product Length (bp)
MEG3	F: CAGGACGACCAAGGAGGAGGAC	60	135
	R: TCAGAGCAACAAGGCAGAAGCATAG		
*FASLG*	F: CCTGTGTCTCCTTGTGATGT	57	165
	R: TTGGGGTGACCTATTTGCTT		
*TNF*	F: GCACTGAGAGCATGATCCG	57	161
	R: AACCTCGAAGTGCAGTAGG		
*BAX*	F: TGAGCAGATCATGAAGACAGGG	58	142
	R: GAGACACTCGCTCAACTTCTT		
*BCL2*	F: GGCCTTCTTTGAGTTCGGT	58	163
	R: ATACAGCTCCACAAAGGCATC		
*CASP8*	F: TCTGAGCAAGACCTTTAGTG	55	107
	R: TATGGTCCAAGTTTCGGTAG		
*CASP9*	F: AAGCAAATGGTCCAGGCTTT	57	164
	R: ACAATTTTCTCCACGGACAC		
*CASP3*	F: ACAGCACCTGGTTACTATTC	55	145
	R: ATTCTACTGCTACCTTTCGG		
*CYCS*	F: GATTCACTTACACAGATGCCA	56	102
	R: TTGTTCCAGGGATGTACTTCTT		
*TP53*	F: GATGAAAATCCAGATGACGC	55	150
	R: TAGACGGAAATCATAGCTGC		
*AI* *FM1*	F: TAGAACTCCAGATGACAAGAC	54	100
	R: CCTATTGTTGATAAGCCCAC		
*ENDOG*	F: GAGCCGCGAGTCTTATGT	57	121
	R: ATGGAAGTCACAAGAGCGG		
ssc-miR-135	F: TGCGGCGTATGGCTTTTTATTCCTATG	60	-
	R: AGTGCAGGGTCCGAGGTATT		
*U6*	F: CTCGCTTCGGCAGCACA	60	-
	R: AACGCTTCACGAATTTGCGT		
*GAPDH*	F: CACAGTCAAGGCGGAGAAC	58	106
	R: CGTAGCACCAGCATCACC		

## Data Availability

The original contributions presented in this study are included in the article/Appendix A. Further inquiries can be directed to the corresponding author.

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
