# Peer review of "LncRNA MEG3 Regulates *Glaesserella parasuis*-Induced Apoptosis of Porcine Alveolar Macrophages via Regulating ssc-miR-135/CASP8 Axis"

_microorganisms, 2025, doi:10.3390/microorganisms13102287_

Round 1
Reviewer 1 Report
Comments and Suggestions for Authors
This paper presents the results of a study on the molecular basis of apoptosis induction by the bacterium Glaesserella parasuis in porcine alveolar macrophages. The work involves the expression of genes whose products are essential for the extrinsic pathway of apoptosis, and their relationship with various regulatory RNAs.
The materials and methodology are adequate and thoroughly described, and the results are clearly presented, accompanied by a large number of graphs and images. However, further work is required for publication; it is especially important to delve into aspects that allow us to explain the mechanisms that could generate these changes. The authors mention in the introduction that apoptosis induction is regulated by virulence factors that they never describe. It is necessary to describe the pathogenic mechanism of the bacterium, listing these virulence factors that could be involved in the death of infected cells, and then analyze them in relation to the results in the discussion section.
Some minor issues that need to be corrected are noted below.
Line 40: The concept of "important" is very general and relative in this case; please delete the term.
In the discussion, rewriting the first sentence, it is confusing.
The statement about the percentage of transcription attributed to the reference.
Lindberg, J.; Lundeberg, J. The plasticity of the mammalian transcriptome. Genomics 2010,
95, 1-6., is not correct.
Author Response
Comments 1: The materials and methodology are adequate and thoroughly described, and the results are clearly presented, accompanied by a large number of graphs and images. However, further work is required for publication; it is especially important to delve into aspects that allow us to explain the mechanisms that could generate these changes. The authors mention in the introduction that apoptosis induction is regulated by virulence factors that they never describe. It is necessary to describe the pathogenic mechanism of the bacterium, listing these virulence factors that could be involved in the death of infected cells, and then analyze them in relation to the results in the discussion section. |
Response 1: Thank you very much for your recognition of our manuscript and for the valuable suggestions you provided. Based on your suggestions, we have made additions to the discussion section and have discussed it alongside the results, hoping to gain your approval (Please see: lines 407-410). |
Comments 2: Line 40: The concept of "important" is very general and relative in this case; please delete the term. |
Response 2: Thank you very much for your professional advice. Based on your suggestion, we have made it deletions in the manuscript (Please see: line 40). Comments 3: In the discussion, rewriting the first sentence, it is confusing. Response 3: We sincerely apologize for any confusion caused by our unclear description. Based on your suggestions, we have made revisions and hope to gain your approval. Thank you once again (Please see: lines 393-394). Comments 4: The statement about the percentage of transcription attributed to the reference. Lindberg, J.; Lundeberg, J. The plasticity of the mammalian transcriptome. Genomics 2010, 95, 1-6., is not correct. Response 4: We sincerely apologize for the inaccurate description caused by our oversight. In order to discuss the content better, we have made modifications to this part and hope to receive your approval. Once again, we would like to thank you for your efforts in facilitating the successful publication of our manuscript (Please see: lines 393-412). |

Reviewer 2 Report
Comments and Suggestions for Authors
The objective of this study was to investigate the role of MEG3 in G. parasuis-induced apoptosis of the porcine 3D4/21 cell line. Consequently, the authors successfully provide an insights into the pathogenic molecular mechanism of G. parasuis, revealing molecular mechanisms of G. parasuis-induced apoptosis via MEG3/MIR135/CASP8.
I recommend this manuscript for publication in Journal Microorganisms, after revision regarding the points shown below:
1. The writing style of genes and/or proteins in Table 1 should be unified, as official gene symbol for example.
2. Letters used in all the figure panels are too small to understand the results, which needs revision with enlargement of the letters and/or the figures themselves.
3. line 292: Methods regarding GO and KEGG pathway analyses are not shown. The databases and/or software used here need to be described, including the criteria employed here.
4. Large part of Discussion, particularly the former half, is written as introduction. Please reconsider reduction of the volume or moving and integrating with Introduction.
Author Response
Comments 1: The writing style of genes and/or proteins in Table 1 should be unified, as official gene symbol for example. |
Response 1: Thank you very much for your recognition of our manuscript, and we also appreciate your efforts in facilitating the smooth publication of our manuscript. Based on your suggestions, we have made changes in Table 1 and hope to receive your approval (Please see: Table 1). |
Comments 2: Letters used in all the figure panels are too small to understand the results, which needs revision with enlargement of the letters and/or the figures themselves. |
Response 2: Thank you very much for your professional advice, which is crucial for improving the quality of our manuscript. Unfortunately, there is still no significant difference in the size of the figures presented after we inserted the manuscript, even though we have made larger adjustments to the font size and dimensions of the figures, which has been very troubling for us. Therefore, we have attached all the figures in their original size for your further review, hoping that this format will be acceptable to you. We also appreciate your efforts in helping our manuscript be published successfully (Please see: Additional clarifications). Comments 3: line 292: Methods regarding GO and KEGG pathway analyses are not shown. The databases and/or software used here need to be described, including the criteria employed here. Response 3: Thank you very much for your professional advice. Based on your suggestions, we have supplemented the methods and standards used for GO and KEGG pathway analysis, and we hope to gain your approval (Please see: lines 293-307). Comments 4: Large part of Discussion, particularly the former half, is written as introduction. Please reconsider reduction of the volume or moving and integrating with Introduction. Response 4: Thank you very much for your efforts in the successful publication of our manuscript. Indeed, as you mentioned, there was overlap between the discussion and introduction sections, so we have rephrased the discussion and removed the overlapping parts. We hope to receive your approval (Please see: lines 393-423). |

Round 2
Reviewer 1 Report
Comments and Suggestions for Authors
In the present form the manuscript can be accepted
Author Response
Comments 1: In the present form the manuscript can be accepted. |
Response 1: Thank you very much for your efforts in publishing our manuscript, and thank you for the professional advice you provided during the publication process. We appreciate your recognition, and wish you a happy life. |
Reviewer 2 Report
Comments and Suggestions for Authors
Several points still remain to be revised.
1. The writing style of genes and/or proteins in Table 1 should be unified, as official gene symbol for example. Use CASP3 instead of Caspase-3, for example. Revision are required for most of the genes.
2. Letters used in all the figure panels, particularly those used for axes of graphs, are too small to read, which is not accepted even if using supplement, because those are essential data for this study. I recommend rearrangement of the enlarged panels to fit within each page.
3. Methods for GO and KEGG pathway analyses described in line 300-306 should be moved to Materials and Methods section. Further, you should add the information for software/database you used in the analyses, including the criteria employed here.
Author Response
Comments 1: The writing style of genes and/or proteins in Table 1 should be unified, as official gene symbol for example. Use CASP3 instead of Caspase-3, for example. Revision are required for most of the genes. |
Response 1: We sincerely apologize for the additional workload caused by our oversight. Based on your suggestions, we have made revisions to Table 1 and hope to receive your approval (Please see: Table 1). |
Comments 2: Letters used in all the figure panels, particularly those used for axes of graphs, are too small to read, which is not accepted even if using supplement, because those are essential data for this study. I recommend rearrangement of the enlarged panels to fit within each page. |
Response 2: We apologize again for the increase in your workload due to our oversight. Based on your suggestions, we have recreated the figure and enlarged the font size, hoping to gain your approval (Please see: Figure 1-8). Comments 3: Methods for GO and KEGG pathway analyses described in line 300-306 should be moved to Materials and Methods section. Further, you should add the information for software/database you used in the analyses, including the criteria employed here. Response 3: Thank you very much for your professional advice. Based on your suggestions, we have moved the description of this section to the Materials and Methods, and added information about the software/databases used in the analysis, including the standards used here. I hope to receive your approval. Once again, I sincerely thank you (Please see: lines 121-143).
|
